# An Interdisciplinary Approach to Müllerian Outflow Tract Obstruction Associated with Cloacal Malformation and Cloacal Exstrophy

**DOI:** 10.3390/jcm11154408

**Published:** 2022-07-28

**Authors:** Bryan S. Sack, K. Elizabeth Speck, Anastasia L. Hryhorczuk, David E. Sandberg, Kate H. Kraft, Matthew W. Ralls, Catherine E. Keegan, Elisabeth H. Quint, Melina L. Dendrinos

**Affiliations:** 1Department of Urology, Division of Pediatric Urology, University of Michigan, Ann Arbor, MI 48109, USA; kraftk@med.umich.edu; 2Department of Surgery, Section of Pediatric Surgery, University of Michigan, Ann Arbor, MI 48109, USA; speckk@med.umich.edu (K.E.S.); mralls@med.umich.edu (M.W.R.); 3Department of Radiology, University of Michigan, Ann Arbor, MI 48109, USA; ahryhorc@med.umich.edu; 4Department of Pediatrics, Division of Pediatric Psychology, University of Michigan, Ann Arbor, MI 48109, USA; dsandber@med.umich.edu; 5Department of Pediatrics, Division of Pediatric Genetics, Metabolism, and Genomic Medicine, University of Michigan, Ann Arbor, MI 48109, USA; keeganc@med.umich.edu; 6Department of Obstetrics and Gynecology, University of Michigan, Ann Arbor, MI 48109, USA; equint@med.umich.edu (E.H.Q.); mdendrin@med.umich.edu (M.L.D.)

**Keywords:** cloacal malformation, cloacal exstrophy, vaginal obstruction, fertility, congenital anomalies, shared decision making

## Abstract

People with cloacal malformation and 46,XX cloacal exstrophy are at risk of developing Müllerian outflow tract obstruction (OTO). Management of OTO requires expertise of many medical and surgical specialties. The primary presenting symptom associated with OTO is cyclical and later continuous pain and can be initially quelled with hormonal suppression as a temporizing measure to allow for patient maturation. The decision for timing and method of definitive treatment to establish a patent outflow tract that can also be used for penetrative sexual activity and potential fertility is a complicated one and incredibly variable based on patient age alone. To understand the management approach to OTO, we put forth five phases with associated recommendations: (1) caregiver and patient education and evaluation before obstruction; (2) presentation, diagnosis, and symptom temporization; (3) readiness assessment; (4) peri-procedural management; (5) long-term surveillance. This review will emphasize the importance of interdisciplinary team management of the complex shared medical, surgical, and psychological decision making required to successfully guide developing patients with outflow obstruction secondary to cloacal malformations and cloacal exstrophy through adolescence.

## 1. Introduction

Congenital anomalies of the reproductive tract resulting in Müllerian outflow tract obstruction (OTO) are rare and quite varied in terms of anatomy, presentation, and treatment [1], requiring the expertise of many medical and surgical specialties. OTO in patients without associated anomalies of the external genitalia usually presents unexpectedly during adolescence with cyclical or recurrent abdominopelvic pain and primary amenorrhea [2,3]. In individuals with associated early-diagnosed and reconstructed congenital anomalies, such as cloacal malformation and cloacal exstrophy, there is usually awareness by caregivers and healthcare providers of the potential for developing OTO in adolescence. Even with surgical creation of a potential outflow tract in early childhood for later menstrual flow, many of these patients will present with OTO in adolescence [4,5]. The primary presenting symptom associated with OTO is cyclical and later continuous pain and can be initially quelled with hormonal suppression as a temporizing measure to allow for patient maturation [6]. In addition, OTO may allow for neither penetrative sexual activity nor conception, whether spontaneous or assisted.

The decision for timing and method of definitive treatment to establish a patent outflow tract that can also be used for penetrative sexual activity and potential fertility is a complicated one and incredibly variable based on patient age and level of maturity. It requires thorough evaluation of anatomy, likelihood to have or desire future fertility, patient readiness for reconstruction and postoperative dilation if indicated, and, in some instances, gender identity. Family support is crucial to determine how and when to manage the OTO. Management discussions include long-term menstrual suppression, removal or reconstruction of uterine structure(s), and vaginoplasty. During initial evaluation, the healthcare team must be aware of the potential psychological, psychosocial, and psychosexual effects OTO may have had on the patient. This emphasizes the need for a biopsychosocial management strategy [7] rather than strictly a biomedical approach.

Interdisciplinary management is imperative to guide patients through the multifaceted process of treating the OTO. Each specialty within the interdisciplinary team works together jointly with the other disciplines while emphasizing their own specific skillset to create a coherent and encompassing strategy to the approach and management [8] (Table 1). This review will emphasize the importance of interdisciplinary team management of the complex shared medical, surgical, and psychological decision making required to successfully guide developing patients with outflow obstruction secondary to cloacal malformations and cloacal exstrophy through adolescence.

## 2. Congenital Müllerian Anomalies at Risk for Outflow Obstruction

This review will focus on two congenital anomalies that are diagnosed early in life that have an increased risk of developing Müllerian outflow tract obstruction in adolescence.

**Cloacal malformation** has an incidence of 1 in 25,000 live female births. The perineal anatomy includes an imperforate anus with a single opening known as the common channel, which is a confluence of the rectum, vagina(s), and urinary system. These children require urgent fecal diversion with a colostomy and may require a vesicostomy or more often a vaginostomy [9] to allow for adequate urinary drainage, particularly if there is significant hydronephrosis or hydrocolpos. After these temporizing procedures, the infant will undergo separation of the rectal component with anorectoplasty using a posterior sagittal incision or an abdominal approach (open or laparoscopic), typically in the first year of life. Because of the high risk of future OTO after early reconstruction, the decision to perform a vaginoplasty at the time of the anorectoplasty has been questioned. With recent advancements of minimally invasive techniques, some have begun deferring vaginal reconstruction until adolescence because of this risk [10,11].

Vaginoplasty techniques at this age include total or partial urogenital mobilization (i.e., the urinary system and vagina are mobilized to the perineum with or without local skin flapping) or using a large or small bowel interposition to serve as a neovagina. Independent of the technique or decision to perform a vaginoplasty, appropriate delineation of the complete Müllerian tract anatomy is often fraught with difficulty. To highlight the frequency of more complicated anatomy, duplicated proximal vaginas are expected to be identified in 44–63% [5,12]. Furthermore, when surveyed, surgeons were uncertain of the Müllerian anatomy in 13–39% of cases after performing infant reconstruction [5,12]. This anatomical uncertainty, paired with less than desirable functional outcomes, emphasizes the need to rethink the timing of Müllerian reconstruction and highlights the need for longitudinal assessment of these patients for OTO as they enter adolescence.

**Cloacal exstrophy** has an incidence of 1 in nearly 200,000 male and female live births [13]. This anomaly consists of a midline anatomical defect that results in two separate exstrophied bladder plates laterally with a midline exstrophied cecal plate and often intussuscepted distal small bowel. This results in exposed and continuous urinary and fecal drainage. Like patients with cloacal malformation, these children require neonatal fecal diversion with an end colostomy, preserving any available colon. These patients also have an omphalocele, which is typically managed at the time of their fecal diversion. In late infancy, after these temporizing measures, the bladder and lower abdominal wall are closed to create a bladder. This anomaly is also referred to as OEIS complex (omphalocele-exstrophy-imperforate anus-spinal defects) because of the other concomitant anomalies.

Müllerian duct anatomy also varies widely in patients with 46,XX cloacal exstrophy. In one of the largest retrospective series of 30 patients, 43% (13/30) had vaginal and uterine duplication, and of the 47% (14/30) with a single vagina, five had vaginal atresia, and one had lateral displacement [14]. Like those with a cloacal malformation, many of these patients had uncertain anatomy after evaluation in the operating room [14]. For those with identifiable Müllerian anatomy, reconstruction is often pursued at the time of bladder reconstruction in late infancy. This may range from simply incising or removing a vaginal septum between two vaginas to intestinal interposition if the vagina is unable to reach the perineum. If the anatomy appears atretic or too complex for surgical reconstruction, hemi- and complete hysterectomies have been performed during early reconstruction [14,15,16]. However, there are many patients who did not undergo vaginal manipulation at the time of bladder closure, and all of these patients should be monitored closely into and through adolescence.

Although the gastrointestinal and urologic concerns are high priorities early in life, as these children age into adolescence, it is paramount that caregivers, patients, and the primary care providers appreciate the complex and uncertain vaginal and uterine anatomy that may later present with OTO.

## 3. Phases of Müllerian Outflow Tract Obstruction

To understand the different management aspects of outflow tract obstruction, we put forth *five phases* with included management recommendations (Table 2):Caregiver and patient education and evaluation before obstruction;Presentation, diagnosis, and symptom temporization;Readiness assessment;Peri-procedural management;Long-term surveillance.

### 3.1. Caregiver and Patient Education and Evaluation before Obstruction

It is the responsibility of the care team to try and predict future OTO in children with cloacal exstrophy and cloacal malformation before they develop symptoms. This consideration and discussion should begin prior to initial infant reconstruction as part of the shared decision-making process that should include the future risk of OTO. Robust and repeated education should be provided at regular clinic visits for the caregivers and, beginning as early as mid-late childhood, with the patient about the possibility of developing OTO and the associated symptoms. During this pre-obstruction phase, assessment of patient maturity and addressing their emotional needs and potential reactions surrounding the understanding of their diagnosis will help with the readiness assessment at the time of OTO.

Around the time of breast development, even in the absence of OTO symptoms, the reconstructed anatomy should be reviewed to refamiliarize the team of the Müllerian anatomy at the time of initial surgical intervention. Ideally, the surgical team will remain involved in the long-term care given the anatomical variability of each patient. When this is not possible, the past surgical records should be reviewed in detail. It is recommended that about 18 months after thelarche (breast budding) or at least at breast Tanner stage 3, pelvic ultrasounds (US) are obtained every 6 months to evaluate for hematocolpos and/or hematometra, so menstrual suppression can be initiated if either are present [5]. Unfortunately, the OTO is usually diagnosed in a symptomatic state.

### 3.2. Presentation, Diagnosis, and Symptom Temporization

Presentation and diagnosis of OTO is established with clinical history, physical exam, imaging, and an understanding of previous surgical interventions, which is paramount to the eventual understanding of the extent of the outflow obstruction. The symptoms associated with outflow obstruction of menses may initially be nonspecific cyclical or non-cyclical abdominal pain, nausea, and generalized gastrointestinal or urinary symptoms and can easily be confused for appendicitis, ovarian torsion, constipation, passing of a kidney stone, or a urinary tract infection. Alternatively, patients may present with primary amenorrhea or dysmenorrhea with menses. Progressive cyclical pain should raise the suspicion of an outflow obstruction.

Upon presentation, a gentle external genital exam can be considered, dependent upon the patient’s age and comfort level of the patient and their family. Child Life assistance should be considered. External inspection with mild labial retraction can identify the urethra, clitoris, presence of labia, a hymenal opening, and vaginal and rectal openings as well as their reconstructed anatomic relationship to each other. A digital rectal exam, if tolerated, can elucidate a low anterior bulge consistent with an outflow tract obstruction.

Adolescents with a suspected OTO should receive a transabdominal US as their initial imaging evaluation. US is a readily available and well-tolerated imaging modality that avoids both sedation and ionizing radiation. When an OTO is present, the US will demonstrate a distended vagina and/or uterus with low-level internal echoes compatible with blood products. The uterus or uteri can be identified superiorly with or without a cervical impression at the upper aspect of the vagina. Hematosalpinx, or blood-filled fallopian tubes, may also be appreciated, as the blood can pass retrograde. Often, the obstruction is large or complex, and US may not be able to delineate the obstruction clearly. If US findings are unclear or support OTO, a pelvic MRI is recommended shortly thereafter to further delineate the anatomy. An MRI provides excellent tissue characterization and allows for definitive characterization of blood products, which demonstrate hyperintense signal on T1-weighted sequences. It will also better define uterine and vaginal anatomy and their relationship with bowel neovagina if present. The radiological studies should be discussed with the radiologists and care team in detail. These imaging studies should confirm the outflow tract obstruction; however, additional imaging, exam under anesthesia, and endoscopic assessment may be required prior to any further planned interventions or suppression.

While the diagnosis is established, attempts to resolve the patient’s pain using menstrual suppression should be made. Dependent on age and institutional referral patterns, this can be accomplished by the gynecology or pediatric endocrinology teams. Menstrual suppression can be achieved with continuous estrogen-progestin or progestin-only options. In refractory cases, GnRH agonists or antagonists can be considered, often initially in concert with other hormonal suppression or add-back therapy. Complete menstrual suppression is difficult to obtain, and the patient and family should be made aware. Even with successful suppression, the patient’s pain often resolves slowly, and recurrence of pain requires diligent follow-up to identify the underlying cause. Continued pain can be secondary to additional menstrual break through bleeding in an obstructed space [6]; these patients will need re-evaluation. If pain remains an issue, often due to breakthrough bleeding, then continued retrograde menstruation can lead to further endometriosis and affect fertility. The care team should exhaust all possibilities for menstrual suppression, as this provides versatility in management strategies and the ability to delay definitive surgical management until the anatomy is well-defined and the patient feels ready, as determined by the readiness assessment.

### 3.3. Readiness Assessment

Once these adolescents have been diagnosed with an obstruction, and their symptoms have quiesced, the decision regarding if, how, and when to surgically intervene is not algorithmic. Prior to making this complex decision, there are factors that must be considered in this specific patient population. We use shared decision making among the patient, caregivers, and healthcare providers [17]. The primary factors that one must consider when performing the readiness assessment are pain management; desire, ability, and options for future fertility; ability to perform vaginal dilation before or after surgery; and psychological well-being and support system.

To navigate this readiness assessment, it is most beneficial when performed collaboratively in an interdisciplinary setting. We feel that the factors listed above make them the ideal candidates for the interdisciplinary nature of the DSD clinic, as it can address each of these concerns utilizing the expertise from the specific indicated specialists, which may include endocrinology, genetics, gynecology, plastic surgery, psychology, surgery, and urology.

#### Factors to Consider during Readiness Assessment

***Pain Management***—If the pain does not resolve with hormonal suppression, surgical correction may need to be undertaken more urgently prior to optimal patient readiness. Interventional radiology or surgical assistance with percutaneous vaginal or uterine drainage may help with symptom management; however, it is not definitive and has a possible infection risk [18]. Intervening too early with surgical reconstruction has been shown to have an increased failure rate secondary to inability to manage post-op vaginal dilation [10,19,20]. Although menstrual suppression allows for delay in intervention, there may be a longer time to resolution of pain compared to early surgery [6].

***Desire, Ability, and Options for Future Fertility***—Predicting the ability to conceive and carry a pregnancy to term with safe delivery in a person with a history of OTO is complicated. Many patients have already had multiple abdominal operations, leading to adhesions that may affect tubal patency. There are no cases in the current literature that describe successful pregnancy in a case of OTO with the presence of a bowel neovagina. Due to the hematometrocolpos that typically occurs, these patients experience significant retrograde menstruation into the pelvis and have higher frequency of endometriosis, which is another known risk factor for infertility [21,22]. Additionally, assessing cervical competency in these obstructed systems, even with MRI and endoscopy, is challenging. Based on the findings of limited retrospective outcome data from two case series in adult cloacal malformation patients without a history of OTO, 6 of 24 were able to have children (5 by cesarean delivery, 1 vaginally) [12], and 2 of 3 sexually active 46,XX cloacal exstrophy patients were able to conceive [14]. These details help counsel and direct the surgical decision to perform uterine-preserving interventions. Additional important pregnancy considerations include the higher likelihood of preterm labor and the recommendation for a cesarean delivery, which is considerably more complicated because of their variant anatomy and history of multiple pelvic surgeries [23]. The cesarean deliveries are often done with a surgical team approach.

***Ability to Perform Vaginal Dilation***—A major aspect that complicates the outcome and success of vaginal reconstruction is patient participation with vaginal dilation. Stenosis is common after vaginal reconstruction and long-term postoperative dilation is needed to prevent stenosis. The ability to self-dilate the vagina or neovagina requires an understanding of one’s own anatomy, why the procedure was performed, and the necessity for long-term dilation. In early teens, this is physically, emotionally, and psychologically difficult and often results in non-compliance resulting in vaginal stenosis. Thorough counseling with the gynecology, surgery, urology, and psychology teams will hopefully preoperatively predict the inclination and ability to dilate, as many young adolescents are not emotionally ready to perform dilations, and some may not have adequate privacy [24]. Although the patient data are heterogenous in diagnosis, age, and surgical method, the success of vaginoplasty outcomes are improved when they are performed after puberty [10,19,20]. This is believed to stem from compliance with dilation and also estrogenization of the tissue during the post-operative healing process. Many recommend only undertaking these procedures when the patients are ready to initiate vaginal sexual activity [10,19,25]. We feel that the timing for intervention should be informed by the patient’s interests for intimate vaginal activities with or without a partner. However, the prerequisite of being sexually active prior to offering vaginoplasty may be misguided because patients may be avoiding intimate interactions because of their atypical anatomy.

***Psychologic Well-being and Support System***—Determining readiness for a complex surgical reconstruction during the pubertal maturation period requires assessment of the patient’s psychological well-being and the available support around the person. Fortunately, although those with complex anatomies such as cloacal exstrophy and cloacal malformation have other co-morbidities, they are reported to have good psychological functioning [26]. They are also able to socially integrate and adapt as adults [27]; however, adolescents may have anxiety about their genital appearance and sexual activity [27], which further emphasizes the need to inquire about their psychological, psychosocial, and psychosexual thoughts and concerns prior to moving forward with decision making. To help gain an understanding of this, we recommend that they undergo psychological evaluation prior to moving forward with surgical intervention. Utilization of social work support can help provide an understanding of the family and community network that can be used to help with decision making and post-operative care plans. Ensuring a stable support system in the early stages of surgical planning is critical and embarking on these types of interventions independently is discouraged.

At the end of the readiness assessment, the patient should be confident in moving forward with surgery to relieve the OTO. If the patient is not ready to move forward, reassessing periodically (e.g., every 6–12 months or based on symptomatology) is recommended.

These varying factors are intricately woven together and require the skills of many specialties to create a cohesive individualized management strategy. Given the previously illustrated benefits of an interdisciplinary setting [8], such as a DSD or anorectal malformation clinic, these patients are ideal candidates for management within this organizational structure.

### 3.4. Peri-Procedural Management

When the healthcare team and patient have decided to move forward with surgical intervention for the outflow tract obstruction, there are certain anatomic considerations and patient decisions that will inform whether vaginoplasty, uterine reconnection or reconstruction, hysterectomy, or any combination are performed.

***What is necessary for the long-term health and well-being of the patient?*** The presence of OTO and the associated surgical intervention can lead to life-altering and potentially life-threatening infection and/or urinary or gastrointestinal tract complications. For this reason, all of the goals listed below should be balanced against the short- and long-term surgical and medical risks in a comprehensive shared decision-making process.

***What is the goal for sexual activity?*** The ability to have penetrative sexual activity may not be an immediate concern for these patients, as they often present in younger adolescence, or they may prefer different intimate activities. However, if menstrual suppression is the initial management strategy, the desire to become sexually active or have the ability to decide to do so may factor into decisions regarding the timing and type of surgical management. The goal of surgery is to create a vaginal introitus that is both large enough for menstrual egress and functional for penetrative sexual activities, should the patient desire to do so in the future.

***What is the goal for fertility preservation?*** The fertility issue may require multiple counseling sessions and, if needed, involvement of a reproductive endocrinology and infertility (REI) specialist. REI input is helpful, as the patient’s congenital and reconstructed anatomy may not allow for spontaneous conception, oocyte harvesting, embryo implantation, and/or carrying of a pregnancy to term. These considerations will help determine if uterine preservation is helpful to the patient’s goals.

Several retrospective series show that, historically, patient and provider have tended towards hysterectomy to eliminate the risk of recurrent OTO, which is the concern with uterine-preserving procedures. Specific to patients with a cloacal malformation, in one older series of 22 patients, all 9 of the post-pubertal patients who presented with abdominal pain and obstruction underwent a hemi- or total hysterectomy [28]. For patients with cloacal exstrophy, 13 of the 31 individuals in one series underwent Müllerian procedures after initial reconstruction; 7 of these 13 underwent hemi- or total hysterectomy, with OTO listed as one of the indications [14]. A well-described timeline of interventions for patients with 46,XX cloacal exstrophy showed that, of 13 post-pubertal patients, 6 retained their entire uterus, 2 had hemi-hysterectomies, and the remaining 5 had total hysterectomies. Of note, the two patients with OTO both had hysterectomies [15]. Although there may be a possibility for pregnancy in these patients, when weighing the risks and benefits of recurrent OTO and need for re-operation, many patients choose hysterectomy. However, a uterine-preserving procedure should always be discussed if the patient is understanding of the risks and benefits. Clearly, patient age and maturity as well as family dynamics and consent issues are a complex part of these decisions.

Some patients may have questions regarding heritability of this condition when deciding about future fertility. Although there is currently no known genetic etiology of these conditions, and there are not enough offspring reported to see if there is a clinically significant increased risk of heritability, this would be an ideal time to involve the genetics team to help address this concern. Given the rapid pace of identification of genetic conditions with the incorporation of next-generation sequencing methods into clinical testing, it is possible that a genetic etiology might be identified in the future.

***Where is the obstruction?*** After years of menstrual suppression, updated imaging may be necessary to redefine the OTO. Dependent upon their congenital or reconstructed anatomy, the obstruction may be just inside the introitus or more proximal. If it is a proximal obstruction, it could be related to the bowel neovagina anastomosis site or a congenital obstruction if the vagina failed to be identified at the infant evaluation. The location of the obstruction may have been elucidated from a previous MRI; however, vaginoscopy or fluoroscopy can aide in determining the optimal surgical approach.

***What should the surgical approach be?*** Surgical creativity may be necessary when approaching these patients, and it is imperative to discuss with the patients that there are several options and that the final decision may need to be made in the operating room by the surgical team. In order to relieve the obstruction, either the native vagina needs to be brought down to the perineum, or the gap between the introitus and the native vagina or uterus needs to be bridged. There are a variety of described options and numerous considerations to accomplish this goal. Each of these different approaches has varying degrees of invasiveness and surgical risk, which is imperative to discuss and explore with the patient and their support system prior to surgical consent. This clearly highlights the need for interdisciplinary surgical care with pediatric surgery, urology, gynecology, and plastic surgery collaboration. Options for surgical intervention may include:Interposition with a skin graft or buccal mucosa [29];Small or large bowel interposition [30];Complex pediatric laparoscopy to mobilize the native vagina down to the perineum [31];Vaginal switch technique—in the presence of two hemivaginas and two hemiureteri, one hemiuterus is removed, and the associated hemivagina is tubularized with the contralateral hemivagina to lengthen the vaginal canal to the perineum [32].

### 3.5. Long-Term Surveillance

Recurrence of outflow obstruction is predicated on the patient having had a uterine-preserving procedure. Those who have uterine-preserving procedures must undergo regular symptom and intermittent imaging evaluations to ensure recurrence of OTO is identified early. These patients may require continued hormonal suppression to aid in dilation and treatment of possible endometriosis. If there is recurrence of OTO, we recommend repeating the management strategy outlined in this review (Table 2). All patients should follow closely with their reconstruction team and, in particular, gynecology for vaginal patency and sexual activity concerns as well as contraception, suppression for pain, and preventive health screening. If sexuality issues arise, psychology or sexual health therapists could be consulted.

For those interested in pregnancy, we recommend preconception counseling with REI, maternal fetal medicine, and the previous surgical team to create a pregnancy and delivery plan prior to conception. Cesarean delivery is likely to be recommended due to the risk to their reconstructed pelvic floor anatomy that can occur with vaginal delivery in patients with a cloacal malformation or cloacal exstrophy. These patients may have transitioned away from the surgical team; however, most tertiary care facilities will have gynecologic, urologic, and surgical specialists that will help navigate anatomic concerns surrounding pregnancy and delivery. Adult teams should not hesitate to involve their pediatric counterparts even as patients age.

## 4. Conclusions

Adolescent Müllerian outflow tract obstruction in patients born with a cloacal malformation or 46,XX cloacal exstrophy requires the expertise of multiple medical and surgical specialties. We strongly recommend an interdisciplinary team approach to navigate the stages of pre-obstruction education, presentation and symptom temporization, readiness assessment, peri-procedural management, and long-term surveillance. This expanded team approach should improve short- and long-term outcomes and patient satisfaction.

## Figures and Tables

**Table 1 jcm-11-04408-t001:** Medical and Surgical Specialty Involvement at the Different Phases of Müllerian Outflow Tract Obstruction (specialties listed in alphabetical order).

Caregiver and Patient Education and Evaluation before Obstruction	Presentation, Diagnosis, and Symptom Temporization	Readiness Assessment	Peri-Procedural Management	Long-Term Surveillance
Gynecology	Endocrinology	Genetics	Gynecology	Gynecology
Pediatric Surgery	Gynecology	Gynecology	Pediatric Surgery	Maternal Fetal Medicine
Primary Care	Primary Care	Primary Care	Plastic Surgery	Previous Surgical Team
Psychology	Radiology	Psychology	Primary Care	Primary Care
Urology		Social Work	Psychology	Psychology
			Radiology	Reproductive Endocrinology and Infertility
			Social work	
			Urology	

**Table 2 jcm-11-04408-t002:** Management Recommendations at the Different Phases of Müllerian Outflow Tract Obstruction.

Caregiver and Patient Education and Evaluation before Obstruction	Presentation, Diagnosis, and Symptom Temporization	Readiness Assessment	Peri-Procedural Management	Long-Term Surveillance
Shared decision making about potential OTO at initial reconstruction	Symptom assessment	Re-evaluation of pain management	Establish sexual activity goals	Evaluate recurrence of obstruction
Education about potential OTO at regular clinic visits	Ultrasound followed by MRI	Discuss desire and options for future fertility	Establish goals for fertility preservation	Determine need for continued vaginal dilation
Pelvic ultrasounds starting about 18 months after thelarche, every 6 months to evaluate for silent OTO	Pain control with hormonal suppression	Assess ability to perform vaginal dilation	Determine specific location of obstruction (may require additional imaging/endoscopy)	Perform regular PAP smears and contraception counseling
Evaluate past surgical details at thelarche to understand Müllerian anatomy	Evaluate past surgical details	Evaluate psychologic well-being and support system	Referral to genetics to discuss heritability	Assess sexuality and refer to psychology if concerns arise
			Create and move forward with an agreed upon surgical plan	Pre-conception counseling with REI, MFM, and surgical specialists
				Surgical assistance at time of cesarean delivery

OTO, outflow tract obstruction; MRI, magnetic resonance imaging; REI, reproductive endocrinology and infertility; MFM, maternal fetal medicine.

## Data Availability

Not applicable.

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
