# Peer review of "An Interdisciplinary Approach to Müllerian Outflow Tract Obstruction Associated with Cloacal Malformation and Cloacal Exstrophy"

_jcm, 2022, doi:10.3390/jcm11154408_

Round 1

Reviewer 1 Report

The most valuable message of this review is to manage Müllerian outflow tract obstruction (OTO) associated with cloacal malformation and cloacal exstrophy by a multidisciplinary team.

Most reported papers about OTO describe details of surgeries to repair OTO by a single team such as urology, pediatric surgery, and gynecology, however, OTO should be managed by a multidisciplinary team in 5 phases (1. Caregiver and patient education and evaluation before obstruction, 2. Presentation, diagnosis, and symptom temporization, 3. Readiness assessment, 4. Peri-procedural management, and 5. Long-term surveillance).

The authors’ ideas are reasonable and practical from my experience as a pediatric urologist. This review would contribute to readers that manage OTO. The number and level of references are appropriate.

Author Response

The most valuable message of this review is to manage Müllerian outflow tract obstruction (OTO) associated with cloacal malformation and cloacal exstrophy by a multidisciplinary team.

Most reported papers about OTO describe details of surgeries to repair OTO by a single team such as urology, pediatric surgery, and gynecology, however, OTO should be managed by a multidisciplinary team in 5 phases (1. Caregiver and patient education and evaluation before obstruction, 2. Presentation, diagnosis, and symptom temporization, 3. Readiness assessment, 4. Peri-procedural management, and 5. Long-term surveillance).

The authors’ ideas are reasonable and practical from my experience as a pediatric urologist. This review would contribute to readers that manage OTO. The number and level of references are appropriate.

Response: Thank you Reviewer 1 for the supportive comments.

Reviewer 2 Report

This is an admirable attempt to bring ARM to the attention of the medical providers as a complex health condition requiring a multidisciplinary approach. 

Few comments: Line 95-96: "duplicated proximal vaginas are expected to be 95 identified in 44-63%" - could authors provide the reference? 

Line 221-223: "Although those with cloacal malformation and cloacal exstrophy do not have a “classic” Difference of Sex Development (DSD) because of their lack of known chromosomal or genetic anomalies..."

Recommend to change this statement as at least up to 10% of the ARMs are caused by chromosomal variation: see Moore, Pediatr Surgery Inter, 2013, Marcelis, Am J Med Genetic, 2011. Also, some cases are likely due to the chromosomal variations not yet discovered. All in all, this makes DSD programs a perfect set up for care of these patients. 

Line 231-232: "Interventional radiology or surgical assistance with percutaneous vaginal or uterine drainage may help with symptom management; however, it is not definitive and has a theoretical infection risk" - please, illustrate what literature has been published on this topic (drainage of the hematocolpos) as this is not a decision that should be taken lightly (due to risk of infection) - see Mizia K et al, Müllerian dysgenesis: a review of recent outcomes at Royal Hospital for Women. Aust N Z J Obstet Gynaecol. 2006. This experience was related to Mullerian agenesis but nevertheless. I would recommend changing "theoretical risk" to "possible risk" as we do not have incidence rate due to rarity of the condition. 

  1.  

Author Response

Comment 1: This is an admirable attempt to bring ARM to the attention of the medical providers as a complex health condition requiring a multidisciplinary approach. 

Response 1: Thank you Reviewer 2 for the supportive comment.

Comment 2: Line 95-96: "duplicated proximal vaginas are expected to be 95 identified in 44-63%" - could authors provide the reference? 

Response 2: The 44% estimate is from [Harden, 1998] and the 63% estimate from [Pradhan, 2018]. Citations for these two manuscripts have been added after this sentence.

Comment 3: Line 221-223: "Although those with cloacal malformation and cloacal exstrophy do not have a “classic” Difference of Sex Development (DSD) because of their lack of known chromosomal or genetic anomalies..." Recommend to change this statement as at least up to 10% of the ARMs are caused by chromosomal variation: see Moore, Pediatr Surgery Inter, 2013, Marcelis, Am J Med Genetic, 2011. Also, some cases are likely due to the chromosomal variations not yet discovered. All in all, this makes DSD programs a perfect set up for care of these patients. 

Response 3: Thank you for these citations and for making this point. We agree with this reviewer and our use of “classic” may be confusing and detracting from potential chromosomal variation as the etiology of disease and downplays the rationale for why these patients would be ideal to be a part of the DSD clinic. We think this point would be best emphasized without confusion by simply removing the sentence.

Comment 4: Line 231-232: "Interventional radiology or surgical assistance with percutaneous vaginal or uterine drainage may help with symptom management; however, it is not definitive and has a theoretical infection risk" - please, illustrate what literature has been published on this topic (drainage of the hematocolpos) as this is not a decision that should be taken lightly (due to risk of infection) - see Mizia K et al, Müllerian dysgenesis: a review of recent outcomes at Royal Hospital for Women. Aust N Z J Obstet Gynaecol. 2006. This experience was related to Mullerian agenesis but nevertheless. I would recommend changing "theoretical risk" to "possible risk" as we do not have incidence rate due to rarity of the condition. 

Response 4: Thank you for sharing this citation to help demonstrate the reality of this risk. This citation has been included and “has a theoretical infection risk” has been changed to “has a possible infection risk [Mizia, 2006].”